# Progression of Metabolic Syndrome Components along with Depression Symptoms and High Sensitivity C-Reactive Protein: The Bogalusa Heart Study

**DOI:** 10.3390/ijerph18095010

**Published:** 2021-05-09

**Authors:** Azad R. Bhuiyan, Marinelle Payton, Amal K. Mitra, Sophia S. Leggett, Jihua Xu, Paul B. Tchounwou, Frank Smart

**Affiliations:** 1School of Public Health, College of Health Sciences, Jackson State University, Jackson, MS 39213, USA; marinelle.payton@jsums.edu (M.P.); amal.k.mitra@jsums.edu (A.K.M.); sophia.s.leggett@jsums.edu (S.S.L.); 2Louisiana Health Sciences Center, New Orleans, LA 70112, USA; jxu5@lsuhsc.edu (J.X.); fsmart@lsuhsc.edu (F.S.); 3College of Science, Engineering, and Technology, Jackson State University, Jackson, MS 39217, USA; paul.b.tchounwou@jsums.edu

**Keywords:** metabolic syndrome, central obesity, depression, high sensitivity C-reactive protein, intracellular adhesion molecule-1, Bogalusa heart study

## Abstract

This study examined the association between depression symptoms and metabolic syndrome (MetS) or its components prospectively. It assessed the mediator role of high-sensitivity C-reactive protein (hs-CRP) and intracellular adhesion molecule-1 (ICAM-1). Self-reported depression symptoms were assessed using the Center for Epidemiologic Studies-Depression scale. MetS was defined as having at least three of the following five criteria: (1) waist circumference >102 centimeters (cm) in men or >88 cm in women; (2) triglycerides ≥ 50 milligrams per deciliter (mg/dL); (3) high-density lipoprotein cholesterol <40 mg/dL in men or <50 mg/dL in women; (4) blood pressure: systolic ≥ 30 and diastolic ≥85 mm of mercury or on antihypertensive medication; and (5) fasting glucose ≥110 mg/dL. The risk ratios (RR) with 95% confidence interval (CI) were estimated using multivariate Poisson regression models. A total of 419 White and 180 Black individuals with a mean age of 36 years were followed for 6.9 years. The findings demonstrated that hs-CRP mediated the influence of depression symptoms on central obesity in White young adults. The adjusted RR for central obesity was 1.08 with 95% CI of 0.88–1.32, and the value for hs-CRP was 1.12 with 95% CI of 1.02–1.23. Although depression did not influence MetS in this study cohort, the complete mediator role of hs-CRP was established for central obesity, a component of MetS in White young adults.

## 1. Introduction

Metabolic syndrome (MetS) is a complex disorder that is defined as a cluster of abdominal obesity, hyperglycemia, hypertension, and dyslipidemia [1,2]. According to the National Cholesterol Education Program in Adults [3], MetS is defined as having three of the following five criteria: (1) waist circumference >102 centimeters (cm) in men or >88 cm in women; (2) triglycerides ≥150 milligrams per deciliter (mg/dL); (3) high-density lipoprotein cholesterol (HDL-C) <40 mg/dL in men or <50 mg/dL in women; (4) blood pressure (BP) ≥130/≥85 mm Hg or on antihypertensive medication; and (5) fasting glucose ≥110 mg/dL. The impact of MetS was twofold on cardiovascular disease (CVD) and 1.5-fold on all-cause mortality. The study was based on a systemic review and meta-analysis of 87 prospective epidemiologic studies with 951,083 participants [4]. Similarly, the impact of MetS on the incidence of diabetes was 3.5-fold in a meta-analysis of 16 prospective studies with 42,419 participants [5]. According to the National Health and Nutrition Examination Survey (NHANES), one-third (33.0%) of the U.S. population met the diagnostic criteria for MetS during the years 2003–2012, with a higher prevalence of MetS in White individuals compared to Black individuals (32.7%) [6]. The racial/ethnic differences are explained by an interaction between environmental and genetic factors [7,8].

Although the pathophysiological cause of MetS is still controversial, the central role of abdominal obesity in MetS is well documented [8]. Obesity is a disease condition that is characterized by a low-grade systemic inflammation [9]. In addition to its role in obesity, systemic inflammation, measured by high-sensitivity C-reactive protein (hs-CRP), was associated with MetS in the NHANES Survey [10]. Depression is also a significant public health concern. In the United States, one in five adults has at least one episode of lifetime major depression [11]. Depression is a robust psychosocial predictor of CVD and type-2 diabetes [12,13].

Beyond the role of obesity and hs-CRP, several studies stated that depression is linked to MetS or its components through a biological mechanism that involves the hypothalamic-pituitary-adrenal (HPA)-axis [14]. Recently, cross-sectional studies have indicated that depression is related to MetS through inflammatory biomarkers [15]. However, limited longitudinal studies have suggested that depression predicts MetS, but the role of inflammatory biomarkers is not clear [16,17]. Inflammatory biomarkers, such as interleukin-6, plasma intracellular adhesion molecule-1 (ICAM-1), and hs-CRP, promote inflammation and play a role in the pathogenesis of atherosclerotic CVD. The epidemiological follow-up study reported that both ICAM-1 and hs-CRP were associated with CVD. Participants with hs-CRP documented elevated coronary risk if ICAM-1 was high [18]. hs-CRP has emerged as an independent predictor of CVD and type-2 diabetes [19,20]. Consequently, the American Heart Association and the Centers for Disease Control and Prevention (CDC) have recommended guidelines for incorporating hs-CRP into CVD risk stratification [19].

Bidirectional and conflicting results have been reported on the potential association between depression and MetS and vice versa [21,22,23]. Co-occurrence of depression and MetS also have been documented, and MetS might be the hallmark of the unhealthy lifestyle habits of depression [24]. Moreover, the role of inflammatory biomarkers was not apparent in previous studies [16,17]. Given the high prevalence of major depression and MetS in the population, we hypothesized that depression symptoms influence metabolic syndrome or its components through the action of inflammatory biomarkers such as hs-CRP and ICAM-1. During the follow-up period of asymptomatic young adults in 6.9 years, our specific aims were to (1) prospectively examine the association between depression symptoms and MetS or its components as outcomes, and (2) investigate the mediator role of hs-CRP and ICAM-1 on their association.

## 2. Materials and Methods

### 2.1. Study Population

This prospective cohort study was derived from two cross-sectional surveys conducted in the Bogalusa Heart Study (BHS). Bogalusa is situated in Washington Parish, 60 miles northeast of New Orleans, Louisiana. The 2020 US census recorded a population of 11,202 with an average income of USD 45,004 and a poverty rate of 31.53%, ranking 32nd in the state. The BHS is a longitudinal epidemiological study of the natural history of CVD in children and young adults of a rural population in Bogalusa, LA. Initially, the study was funded by the National Institutes of Health (NIH) for CVD epidemiology research in 1972 [25]. During the 2001–2002 survey, young Black and White adults aged 18 to 45 years (*n* = 1203) residing in Bogalusa, LA, were examined as part of the long-term cohort follow-up study. Bogalusa’s population is composed of 65% White and 35% Black residents. Of these, 797 who had data on depression symptoms, measurements of hs-CRP and ICAM-1 biomarkers, and CVD risk factor variables were included as part of a longitudinal cohort study. During the 2008–2009 survey, the same participants were assessed for CVD risk factor variables. This assessment resulted in 599 young adult Black and White participants from the BHS in the 6.9 year follow-up study.

### 2.2. Ethical Procedures

BHS participants were informed of the aim of the study and the study procedures. An informed consent was obtained from all participants. The Jackson State University Institutional Review Board (IRB) reviewed and approved the secondary data analysis project. The IRB was exempted by Jackson State University as primary data collection was not conducted during this study. Deidentified data were used in this research.

### 2.3. Eligibility Criteria

Participants who were free from CVD and type-2 diabetes were included in the analysis. Participants having MetS at the beginning of the study (132 White and 66 Black subjects) were excluded. The study population was composed of 70% White adults and 42% males.

### 2.4. General Measurements

Trained field observers used standardized techniques and protocols to perform anthropometric and blood pressure measurements. Replicate measurements of the waist were made twice, and the mean values were recorded. Systolic and diastolic blood pressures were measured three times by each of two randomly assigned observers using a mercury sphygmomanometer. The lifestyles variable, based on, e.g., smoking habits and alcohol consumption, was also assessed during the survey. Smokers were defined as those who smoked at least one cigarette per week during the past one year or more [26].

### 2.5. Depression Symptoms (DS)

DS were assessed using the Center for Epidemiological Studies-Depression scale (CES-D) developed by Laurie Radloff in 1977 [27]. The CES-D obtained from the self-reported questionnaire was widely used in psychiatric epidemiologic studies in measuring depression. The CES-D consisted of 20 questions with 4-point scales. Respondents were asked to answer 20 questions about how often they had experienced certain feelings of symptoms during the previous week before they were screened. Responses were recorded on a 4-point scale from rarely (0) to almost always (3). The criterion validity for depression using a diagnostic interview scale was very satisfactory [28]. The cutoff score of 16 or above was considered DS, as recommended for the general population [29].

### 2.6. Laboratory Measurements

Participants were asked to fast for 12 h before coming to the BHS center for screening. A trained field officer checked for compliance during interviews on the morning of screening. Serum total cholesterol and triglycerides were assayed on fasting samples using an enzymatic procedure done with the Hitachi 902 Automatic Analyzer (Roche Diagnostics, Indianapolis, IN, USA). Serum lipoprotein cholesterol levels were analyzed using a combination of heparin-calcium precipitation and agar-agarose gel electrophoresis procedures. The laboratory data were checked for precision and accuracy of lipid measurements using the Centers for Disease Control and Prevention (Atlanta, GA, USA)’s surveillance program. Plasma glucose levels were measured by a glucose oxidase method as part of a multiple chemistry profile (SMA20, Laboratory Corporation of America, Burlington, NC, USA). Plasma hs-CRP levels were measured by latex particle-enhanced immunoturbidimetric assay on a Hitachi 902 Automatic Analyzer (Roche Diagnostics, Indianapolis, IN, USA). The reproducibility of hs-CRP measurement was evaluated based on 10% randomly assigned pairs of blind duplicate analyses, and the intra-class correlation coefficient was 0.99 for hs-CRP. Plasma ICAM-1 was measured by sandwich enzyme immunoassay (R&D Systems, Minneapolis, MN, USA).

### 2.7. Statistical Analysis

Statistical analyses were performed using the SAS system (SAS Institute, Cary, NC, USA) Continuous variables were tested for normality using a Kolmogorov–Smirnov test, and log transformation was applied to approach normality. For non-normally distributed variables, a nonparametric test was applied based on the Mann–Whitney U test. The two-sided *p* values for both parametric and nonparametric tests were reported. A two-sided *p* value < 0.05 was considered statistically significant. All analyses were performed on transformed data where appropriate. Chi-square tests for categorical variables and independent sample *t*-tests for continuous variables were used. A log-binomial/modified Poisson regression model was used to examine whether DS at baseline were associated with developing MetS and or its components over a 6.9 year follow-up period in the Black/White young adult population. The simple log-binomial model was used to select variables associated with MetS and or its components. Variables that were significant at the *p* value of ≤ 0.05 in the simple log-binomial model were entered in the multivariate log-binomial model. The modified Poisson model was applied if convergence failure was noted in the binomial model. The risk ratio (RR) with a 95% confidence interval (CI) was estimated for MetS and its components using the GENMOD procedure. Several regression models were applied to test for mediation of hs-CRP on the path of DS and abdominal obesity [30]. In model 1, simple log-binomial Poisson regression was used to predict abdominal obesity with DS. In model 2, a simple linear regression model was applied to predict hs-CRP with DS. In model 3, simple log-binomial Poisson regression was used to predict abdominal obesity with log hs-CRP. In model 4, multivariate log-binomial Poisson regression was used to predict abdominal obesity controlling for covariates. We also performed a formal test of mediation outlined by Sobel to determine the mediator tole of hs-CRP on abdominal obesity [31]. Sensitivity and specificity of predicting abdominal obesity also were provided in the receiver operator characteristics (ROC) curve, showing the area under the curve.

## 3. Results

### 3.1. Baseline Characteristics

Table 1 shows the baseline characteristics of the study population. The mean age was 36 years in both races. Twenty-two percent of Black participants had completed a college education, whereas 42% of White participants completed a college education (*p* < 0.001). Regarding income, 11.3% of Black participants had an annual income >USD 45,000, whereas 54% of White participants had an income > USD 45,000 (*p* < 0.001). Physical activity was similar in both races, with 51% Black vs. 60% White adults moderately active. Black participants had significantly higher body mass index (BMI) compared to White participants (29.6 ± 7.4, 26.9 ± 5.4, *p* < 0.001). Waist circumference, systolic and diastolic BP, and insulin levels were significantly higher (*p* < 0.001) among Black than White participants. The median values of insulin and homeostasis model assessment of insulin resistance (HOMA-IR) were higher among Black compared to White participants (*p* < 0.05). However, LDL cholesterol and triglycerides were significantly lower among Black than White participants (*p* < 0.001). HDL cholesterol was significantly higher among Black participants (*p* < 0.001). The prevalence of smoking was 37% and 30% in Black and White participants (*p* = 0.09), respectively. The median value of hs-CRP was higher in Black compared to White participants (*p* = 0.02). On the contrary, there was no difference in ICAM-1 values between races (*p* = 0.09). Alcohol drinking prevalence was the same for Black and White participants (64.2% vs. 63.3%, respectively, *p* = 0.83).

### 3.2. Predictors of Metabolic Syndrome by Race

Table 2 shows the development of MetS by race. Low income, drinking alcohol, high CVD risk factors, and hs-CRP were significant predictors for developing MetS in White participants. CVD risk factors, smoking, and median value of hs-CRP were substantial predictors for developing MetS in Black participants. The median values of ICAM-1 were not a significant predictor for developing MetS in both races.

Figure 1 shows that there is no development of MetS and its components by DS status. The incidence of MetS was 22.8% among non-depressed vs. 27.7% among depressed individuals among White adults. This association was nonsignificant as the *p* value was 0.25.

The incidence of high waist circumference was 39.4% among non-depressed and 51.3% among depressed White individuals with a significant *p* value < 0.02. All other MetS components were not significantly associated with DS with *p* value > 0.05.

Figure 2 displays the incidence of MetS and its of components by DS status among Black participants. The incidence of MetS was 39.6% among non-depressed vs. 26.5% among depressed individuals among Black adults. This association was nonsignificant as the *p* value was 0.09. Table 3 shows the crude and adjusted RR of DS and hs-CRP for developing MetS in White and Black participants. In the adjusted model, the RR of MetS was 1.15 (95% CI = 0.82–162) in White and 0.80 (95% CI = 0.49–1.12) in Black participants. As confidence interval contained one, DS did not influence MetS in either White or Black participants, controlling for CVD risk factors.

Table 4 displays the development of central obesity with DS individuals in White and Black participants. In model 1, the RR of abdominal obesity with DS was 1.30, 95% CI = 1.04–1.62; with hs-CRP was 1.31, 95% CI = 1.21–1.43; and with ICAM-1 was 0.82, 95% CI = 0.60–1.11 for predicting central obesity in White adults. In model 2, the RR of abdominal obesity attenuated with DS to 1.14 with 95% CI = 0.92–1.42, and the RR of hs-CRP became stronger, 1.30 with 95% CI = 1.19–1.42. In model 3, after controlling for CVD risk factors, the RR of DS became 1.08 with 95% CI = 0.88–1.32, and the RR of hs-CRP was 1.12 with 95% CI = 1.02–1.23 for predicting central obesity. The mediator role established in central obesity as RR of DS became insignificant after the inclusion of hs-CRP and CVD risk factors in the model in White participants. However, DS did not influence central obesity in Black participants.

### 3.3. Mediator Role of hs-CRP on Abdominal Obesity

The mediator role of hs-CRP is displayed graphically in Figure 3. DS influenced abdominal obesity (β = 0.26, *p* = 0.02) as shown in model 1. In model 2, DS was significantly associated with log hs-CRP (β = 0.42, *p* = 0.005). In model 3, log hs-CRP was significantly associated with abdominal obesity (β = 0.27, *p* < 0.001). In model 4, both DS and log hs-CRP were simultaneously included in the model. The effect of DS on abdominal obesity disappeared (β = 0.08, *p* = 0.45) after adjustment for CVD risk factors, which suggests that the effect of DS on abdominal obesity was completely mediated by hs-CRP. The Sobel test (*z* = 3.11, SE = 0.04, *p* = 0.04) was also performed using model 2 and model 3. It indicated that the effect of DS on abdominal obesity was mediated by hs-CRP. In Black participants, there was no evidence of the mediator role of hs-CRP on abdominal obesity.

Figure 4 displays the area under the curve (0.84) in predicting abdominal obesity, adjusted for hs-CRP, BMI, waist circumference, and HOMA-IR in White participants.

## 4. Discussion

In this young cohort follow-up study, DS was found to effect central obesity, which has gained increasing attention as the core manifestation of MetS. Currently, NCEP: ATP III and International Diabetes Federation criteria are widely used, and both focus on waist circumference as a surrogate measure of central obesity [2]. This finding is consistent with a large five-year follow-up study of White and Black older persons where depression was measured with CES-D scale, and there was an increase in visceral fat measured by computed tomography [32]. These findings may reiterate why depression increased the risk of CVD and diabetes. In our study, we propose a specific pathophysiologic mechanism that links depression to abdominal obesity. With regards to MetS, our findings were contrary to the previous study’s findings, which focused on DS predicting the incidence of MetS among middle-aged U.S. women. In contrast, this study included all young adults [16]. Moreover, previous systemic and review studies noted bidirectional associations that we could not establish in this young cohort study [21,22,23].

This study also evaluated the mediator role of hs-CRP. It sought to determine whether hs-CRP played a role between DS and MetS or its components after controlling for CVD risk factors. We found that DS influenced central obesity measured by waist circumference in White adults. This finding is consistent with results from previous longitudinal studies [15]. Although Black participants had significantly higher abdominal obesity than White participants, DS did not influence central obesity in Black participants. A similar finding was also noted in a previous study, which documented that depression was not related to the MetS in Black participants [33]. In our data, the reliability of internal consistency of the CES-D depression scale measured by Cronbach’s α was 0.78 for White and 0.84 for Black participants. Based on another recommendation [29], CES-D cutoff of 20 instead of 16 also was considered in the analysis, but the results in both situations remained similar. The reason may be partially explained by our use of a self-reported CES-D scale to measure self-reported DS in our study. We further explored the data at baseline metabolic syndrome. At baseline, there was no significant correlation between DS and metabolic syndrome in White participants (*r* = −0.01, *p* = 0.74) or in Black participants (*r* = −0.07, *p* = 0.31). Even before the exclusion of participants, there was no association observed at baseline between DS and MetS in White participants (prevalence of MetS for no-DS vs. DS: 25% vs. 23.5%, *p* = 0.74), as well as in Black participants (prevalence for no-DS vs. DS: 31.4% vs. 25%, *p* = 0.30). To avoid selection bias, we excluded participants having MetS at the beginning of the study. Although MetS has been defined in several methods, we assessed MetS using the NCEP ATP-III criterion, in which central obesity is an optional component. This definition has been widely used in assessing MetS of the general population in the United States.

The observed mediating role of DS on central obesity can be explained by the biological function of the inflammatory biomarker hs-CRP. Previous studies, including that by the National Cholesterol and Educational Program, have stated that obesity plays a role in the development of MetS [3,34]. Increasing obesity induces the release of chemotactic molecules (e.g., monocyte chemoattractant protein-1) with macrophage infiltration and the release of proinflammatory cytokines, such as TNF-α and IL-6 [34]. These cytokines generate an inflammatory reaction. hs-CRP is an acute phase reactant secreted by the liver. It is upregulated by interleukin-6 synthesized by adipose tissue and hormone cortisol and insulin [35]. Thus, it is likely that DS could lead to MetS through the inflammatory biomarker hs-CRP in White adults. Further studies are needed to explore the mediator role of hs-CRP on obesity by race/ethnicity.

The strength of this study was that it was a cohort study of 6.9 years of follow-up of young adults and representing both Black and White adults. One of the limitations of our study was self-reported DS measured by the CES-D scale. However, this scale is widely used in epidemiological studies [27]. The small number of Black participants may also limit the findings’ power and warrant more extensive investigations. Moreover, our data demonstrated that Black adults had higher BMI, waist circumference, and hs-CRP at baseline.

## 5. Conclusions

Our study results provided clear evidence that DS affects central obesity, a component of MetS [3], and highlighted the mediator role of hs-CRP in White adults. Although, as noted earlier, the Bogalusa population has an average income of USD 45,004 and a poverty rate of 31.53%, Black participants had a lower socio-economic status than White participants in this study. Interestingly, this study showed a significant effect of DS and hs-CRP on central obesity in White individuals. In terms of SES, we did not find an effect of SES on MetS and central obesity in this young cohort in the adjusted model. As noted earlier, depression is a robust psychosocial predictor of CVD and type-2 diabetes [12,13]. Thus, longitudinal studies and clinical trials are needed to delineate the causal relationship and the interplay of hs-CRP between depression and central obesity.

## Figures and Tables

**Figure 1 ijerph-18-05010-f001:**
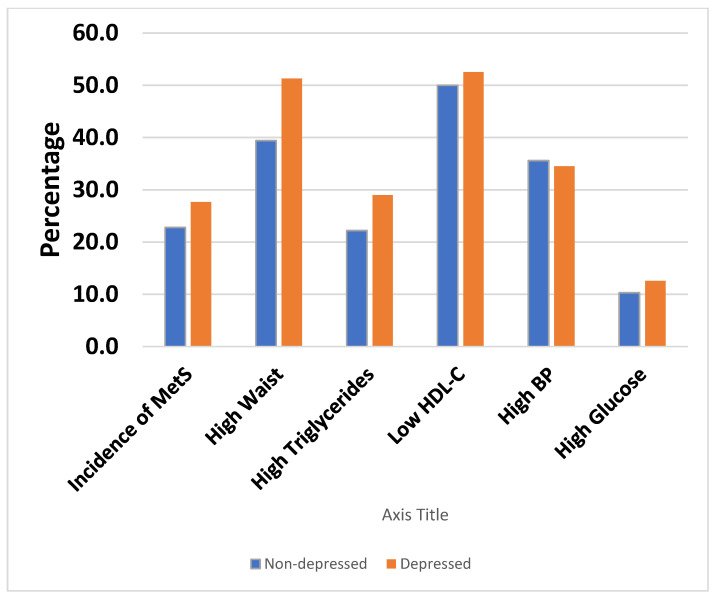
Incidence of MetS and its components by DS among White participants.

**Figure 2 ijerph-18-05010-f002:**
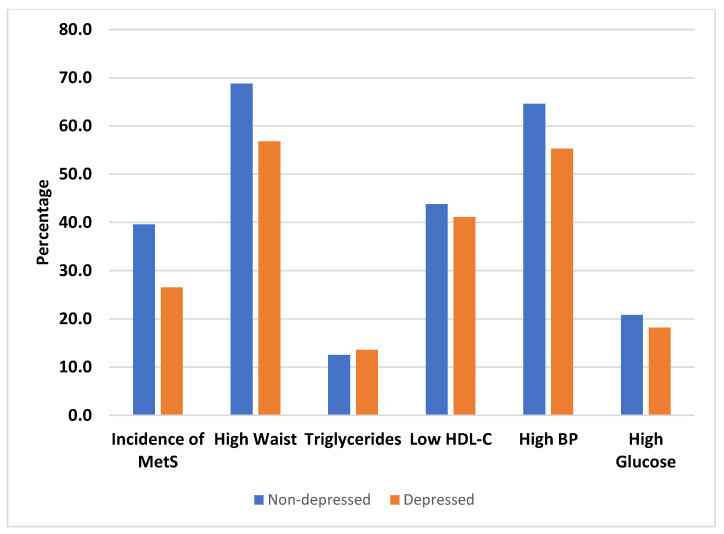
Incidence of MetS and its components by DS among Black participants.

**Figure 3 ijerph-18-05010-f003:**
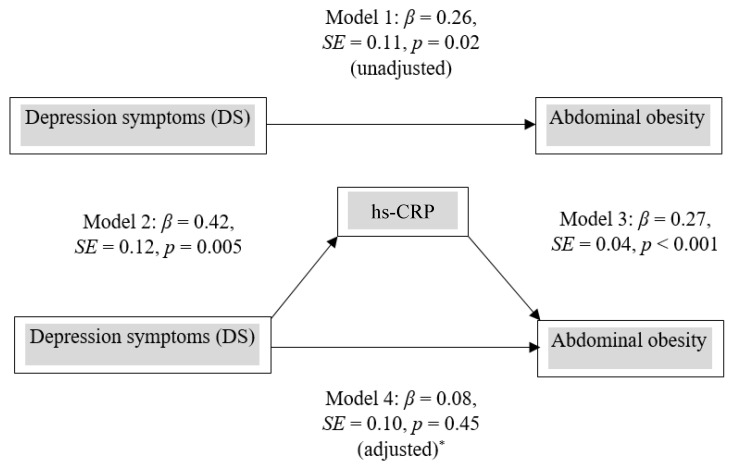
Mediator role of hs-CRP on abdominal obesity in White participants * Adjusted for BMI, waist circumference and HOMA-IR.

**Figure 4 ijerph-18-05010-f004:**
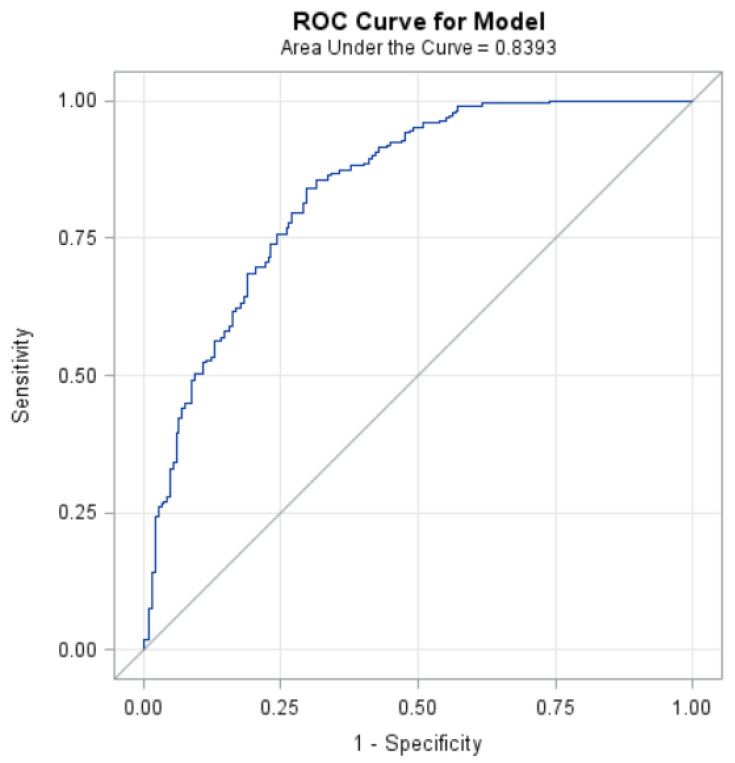
ROC for model predicting abdominal obesity.

**Table 1 ijerph-18-05010-t001:** Baseline characteristics of population: The Bogalusa Heart Study.

Variables	White Participants (*n* = 419)	Black Participants (*n* = 180)	*p* Value ^b^
Age (years) ^a^	36.6 ± 4.3	36.1 ± 4.4	0.17
Education			
Grade 1–12 or GED	128 (31.1%)	109 (61.2%)	<0.001
Vocational/Technical	81 (19.7%)	23 (12.9%)	
College	172 (41.8%)	40 (22.5%)	
Postgraduate	31 (7.5%)	6 (3.4%)	
Income (USD)			
<15,000	42 (10.0%)	96 (54.2%)	<0.001
15,001–29,999	81 (19.4%)	43 (24.3%)	
30,000–45,000	68 (16.3%)	18 (10.2%)	
>45,000	227 (54.3%)	20 (11.3%)	
Physical activity			
No/mild	31 (7%)	23 (13%)	0.05
Moderate	249 (60%)	92 (51%)	
Very active	138 (33)	65 (36%)	
BMI (kg/m^2^) ^a^	26.9 ± 5.4	29.6 ± 7.4	<0.001
Waist circumference (cm) ^a^	87.6 ± 13.9	92.0 ± 16.5	0.002
Systolic BP (mmHg) ^a^	111.4 ± 9.7	119.5 ± 15.1	<0.001
Diastolic BP (mmHg) ^a^	75.5 ± 7.6	80.2 ± 10.7	<0.001
Fasting blood sugar (mg/dL) ^**^ (25th and 75th percentile)	81.0	82.0	0.73
	(76.0–87.0)	(75.0–88.0)	
Insulin (µU/mL) median value ** (25th and 75th percentile)	8.0	10.0	0.005
	(6.0–12.0)	(7.0–17.0)	
* HOMA-IR median value ^**^ (25th and 75th percentile)	0.52	0.71	0.003
	(0.14–0.93)	(0.23–1.27)	0.002
LDL cholesterol (mg/dL) ^a^ HDL cholesterol (mg/dL) ^a^	124.5 ± 32.0	115.9 ± 31.10	<0.001
	48.6 ±13.2	54.3 ± 14.3	<0.001
Triglycerides (mg/dL) median ** (25th and 75th percentile)	99.0	82.0	<0.001
	(72.0–134.0)	(59.0–106.0)	
hs-CRP (mg/L) median value ** (25th and 75th percentile)	1.17	1.82	0.02
	(0.47–2.90)	(0.65–3.62)	
ICAM-1 (ng/mL) median value ** (25th and 75th percentile)	258	253	0.09
	(217–321)	(189–315)	
Smoking prevalence	125 (30.0%)	66 (37.0%)	0.09
Depressed	238 (57.0%)	132 (73.3%)	0.002
Not depressed	180 (43.0%)	48 (26.7%)	
Drinking alcohol for last 12 months			
No	150 (35.8%)	66 (36.7%)	0.83
Yes	269 (64.2%)	114 (63.3%)	0.17

^a^ BMI, body mass index; * homeostasis model assessment of insulin resistance (HOMA-IR) = (Insulin × Glucose × 0.056)/22.5, ^a^ Mean (SD), ^b^
*p* value for race difference. ** Mann–Whitney U test.

**Table 2 ijerph-18-05010-t002:** Association of metabolic syndrome with baseline characteristics: The Bogalusa Heart Study.

	White Participants			Black Participants		
Variables	No MetS (*n* = 312)	MetS (*n* = 107)	*p* Value ^b^	No MetS (*n* = 126)	MetS (*n* = 54)	*p* Value ^b^
Gender			0.18			0.73
Male	123 (39.4%)	50 (46.7%)		43 (34.1%)	17 (31.5%)	
Female	189 (60.6%)	57 (53.3%)		83 (65.9%)	37 (68.5%)	
Age (years) ^a^	36.6 (4.3)	36.6 (4.4)	0.97	36.3 (4.3)	35.5 (4.5)	0.27
Education						
Grade 1–12 or GED	90 (29.4%)	38 (35.9%)		75 (60.5%)	34 (31.2%)	0.81 ^$^
Vocational/Technical	64 (20.9%)	17 (16.0%)	0.3	18 (14.5%)	5 (21.7%)	
College	126 (41.2%)	46 (43.4%)		27 (21.8%)	13 (32.5%)	
Postgraduate	26 (8.5%)	5 (4.7)		4 (3.2%)	2 (3.7%)	
Income (USD)						
<15,000	29 (9.3%)	13 (12.2%)		69 (56.1%)	27 (50.0%)	0.14
15,001–29,999	49 (15.8%)	32 (29.9%)	0.006	33 (26.8%)	10 (18.5%)	
30,000–45,000	52 (16.7%)	16 (15.0%)		9 (7.2%)	9 (16.7%)	
>45,000	181 (58.2%)	46 (43.0%)		12 (9.8%)	8 (14.8%)	
Smoking						
No	220 (71.0%)	72 (67.3%)	0.47	72 (57.6%)	41 (75.9%)	0.02
Yes	72 (29.0%)	35 (32.7%)		53 (42.4%)	13 (24.1%)	
Drinking alcohol						
No drinking	100 (32.0%)	50 (46.7%)	0.006	42 (33.3%)	24 (44.4%)	0.15
Drinking	212 (68.0%)	57 (53.3%)		34 (66.7%	18 (55.6%)	
Physical activity						
No/mild	20 (6.4%)	11 (10.3%)	0.13	19 (15.1%)	4 (7.4%)	0.23
Moderate	181 (58.2%)	68 (63.6%)		60 (47.6%)	32 (59.3%)	
Very active	110 (35.4%)	28 (26.2%)		47 (37.3%)	18 (33.3%)	
BMI (kg/m^2^) ^a^	26.0 (5.2)	29.4 (5.4)	<0.001	28.2 (7.3)	32.9 (6.3)	<0.001
Waist circumference (cm) ^a^	85.2 (13.3)	94.7 (13.3)	<0.001	88.4 (15.8)	100.5 (15.0)	<0.001
Systolic BP (mm Hg) ^a^	110.7 (9.6)	113.5 (9.7)	0.01	116.8 (12.7)	125.8 (12.7)	0.002
Diastolic BP (mm Hg) ^a^	74.9 (7.5)	77.0 (7.6)	0.01	78.6 (10.3)	83.9 (10.6)	0.002
Fasting blood sugar (mg/dL) median * (25th and 75th percentile)	80 (75.0–86.0)	84 (78.0–91.0)	0.001	81 (74.0–87.0)	85 (78.0–98.00)	0.006
Insulin (µU/mL) ^a^ (25th and 75th percentile)	8 (6.0–11.0)	11 (8.0–14.0)	<0.001	9 (6.0–13.0)	15 (9.0–19.0)	<0.001
HOMA-IR (median value) * (25th and 75th percentile)	0.4 (0.08–0.80)	0.87 (0.45–1.10)	<0.001	0.61 (0.15–1.07)	1.15 (0.66–1.44)	0.002
Triglycerides (mg/dL) ^a^ (25th and 75th percentile)	91.5 (67.0–126.0)	115 (89-153)	<0.001	76.5 (53–100)	94 (72–115)	0.004
LDL cholesterol (mg/dl) ^a^	123.0 (31.7)	129.0 (32.4)	0.09	123.0 (31.7)	129.0 (32.4)	0.09
HDL cholesterol (mg/dl) ^a^	50.2 (13.6)	44.1 (10.7)	<0.001	56.5 (14.4)	49.3 (12.6)	0.002
hs-CRP (median value) * (25th and 75th percentile)	1.04 (0.41–2.89)	1.62 (0.81–2.93)	0.005	1.2 (0.44–5.61)	2.89 (1.95–5.44)	<0.001
ICAM-1 median value * (25th and 75th percentile)	255 (213–320)	268 (228–322)	0.42	260 (197–312)	241 (184–322)	0.33

^a^ Mean (SD), ^b^
*p* value for MetS difference; * Mann Whitney U test. ^$^ Fisher’s exact test.

**Table 3 ijerph-18-05010-t003:** Crude and adjusted relative risks of depression symptoms and C-reactive protein for developing metabolic syndrome.

Model	Parameter Estimates	*p* Value	Relative Risk (RR)	95% of RR
White Participants				
Model 1 (unadjusted)				
DS (yes vs. no)	0.20 (0.17)	0.25	1.22	0.87–1.71
C-reactive protein	0.15 (0.07)	0.02	1.18	1.02–1.32
Model 2 (both in the model)				
DS (yes vs. no)	0.14 (0.17)	0.41	1.16	0.83–1.63
C-reactive protein	0.16 (0.06)	0.01	1.17	1.03–1.32
Model 3 (adjusted *)				
DS (yes vs. no)	0.14 (0.17)	0.42	1.15	0.82–1.62
C-reactive protein	0.03 (0.07)	0.67	1.03	0.89–1.19
Black Participants				
Model 1 (unadjusted)				
DS	−0.40 (0.22)	0.08	0.67	0.42–1.05
C-reactive protein	0.41 (0.09	<0.001	1.51	1.25–1.82
Model 2 (both in the model)				
DS	−0.31 (0.22)	0.15	0.73	0.47–1.12
C-reactive protein	0.46 (0.10)	<0.003	1.58	1.32–1.90
Model 3 **				
DS (yes vs. no)	−0.21 (0.25)	0.37	0.80	0.49–1.31
C-reactive protein	0.37 (0.11)	0.005	1.44	1.18–1.78

* Adjusted for income, drinking alcohol, BMI, waist circumference, systolic and diastolic blood pressure, triglycerides, HDL cholesterol, and HOMA-IR. ** Adjusted for gender, income, current smoking status, drinking alcohol, BMI, waist circumference, systolic and diastolic blood pressure, triglycerides, HDL cholesterol, and HOMA-IR.

**Table 4 ijerph-18-05010-t004:** Crude and adjusted relative risks of depression symptoms and C-reactive protein for developing central obesity in White and Black participants: The Bogalusa Heart Study.

Model Selection	Parameter Estimates	*p* Value	Relative Risk (RR)	95% of RR
***White Participants***				
Model 1 (unadjusted)				
DS (yes vs. no)	0.26 (0.11)	<0.02	1.30	1.04–1.62
C-reactive protein	0.27 (0.04)	<0.001	1.31	1.21–1.43
ICAM-1	−0.20 (0.16)	0.20	0.82	0.60–1.11
Model 2 (Both in the model)				
DS (yes vs. no)	0.14 (0.11)	0.22	1.14	0.92–1.42
C-reactive protein	0.26 (0.04)	<0.001	1.30	1.19–1.42
Model 3 (adjusted) *				
DS	0.08 (0.10)	0.44	1.08	0.88–1.32
C-reactive protein	0.12 (0.05)	0.01	1.12	1.02–1.23
***Black Participants***				
Model 1 (unadjusted)				
DS (yes vs. no)	−0.19 (0.12)	0.12	0.82	0.64–1.05
C-reactive protein	0.25 (0.04)	<0.001	1.30	1.17–1.45
ICAM-1	−0.04 (0.10)	0.74	0.97	0.78–1.19
Model 2 (Both in the model)				
DS (yes vs. no)	−0.14 (0.11)	0.23	0.91	0.75–1.10
C-reactive protein	0.25 (0.06)	<0.001	1.14	1.06–1.22
Model 3 (adjusted) **				
DS (yes vs. no)	0.03 (0.12)	0.91	1.01	0.80–1.23
C-reactive protein	0.07 (0.06)	0.19	1.07	0.96–1.20

* Adjusted for BMI, waist circumference and HOMA-IR. ** Adjusted for current smoking status, BMI, waist circumference, LDL and HDL cholesterol, and HOMA-IR.

## Data Availability

The data presented in this study are available on request from the corresponding author.

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
