# Peer review of "Progression of Metabolic Syndrome Components along with Depression Symptoms and High Sensitivity C-Reactive Protein: The Bogalusa Heart Study"

_ijerph, 2021, doi:10.3390/ijerph18095010_

Round 1

Reviewer 1 Report

The authors have put a lot of work into the revisions of this manuscript and it is much improved. However, my main criticisms of the methods have and to some extent cannot be addressed. While the authors tried to address their hypothesis and why they chose the direction of effects they did, ultimately their reasoning is unconvincing: "Although bidirectional and conflicting results have been reported on the potential association between depression and MetS and vice versa [27-29] and role of inflammatory biomarkers were not clear [21,22], we hypothesized that depression symptoms influence metabolic syndrome or its components through the action of inflammatory biomarker, hs-CRP and ICAM-1."

Secondly, the long-term follow-up is good but again the authors have not addressed the fact that they have not followed the participants up more recently. 

Author Response

The authors have put a lot of work into the revisions of this manuscript and it is much improved. However, my main criticisms of the methods have and to some extent cannot be addressed. While the authors tried to address their hypothesis and why they chose the direction of effects they did, ultimately their reasoning is unconvincing: "Although bidirectional and conflicting results have been reported on the potential association between depression and MetS and vice versa [27-29] and role of inflammatory biomarkers were not clear [21,22], we hypothesized that depression symptoms influence metabolic syndrome or its components through the action of inflammatory biomarker, hs-CRP and ICAM-1."

Secondly, the long-term follow-up is good but again the authors have not addressed the fact that they have not followed the participants up more recently. 

Answer: "Although bidirectional and conflicting results have been reported on the potential association between depression and MetS and vice versa [27-29], co-occurrence of depression and MetS also documented and MetS might be the hallmark of unhealthy lifestyle habits of depression. (Marazziti, D. et al). MetS constitutes a major problem of public health that is associated with increased risk of mortality and poor quality of life (4-6). Both conditions trigger a low grade chronic inflammatory state which in turn leads to increased oxidative states of different organ system. According to the National Institute of Mental Health, prevalence of major depression in US is high in 2017. It has reported that 17.3 million adults aged 18 or older in the US had major depression with white individuals experiencing more depression than that of black individuals (7.9% vs 5.4%, respectively).  

References:

Marazziti, D., Rutigliano, G., Baroni, S., Landi, P., & Dell'Osso, L. (2014). Metabolic syndrome and major depression. CNS Spectrums, 19(4), 293-304. doi:10.1017/S1092852913000667

https://www.nimh.nih.gov/health/statistics/major-depression.shtml#part_155029

Wen Y, Liu G, Shang Y, Wang Q. Association of Depression with Metabolic Syndrome in Highly Educated Ethnic Koreans of China: A Case-Control Study. Neuropsychiatr Dis Treat. 2021 Jan 8;17:57-66. doi: 10.2147/NDT.S280716. PMID: 33447037; PMCID: PMC7802915.

Comment 2: Secondly, the long-term follow-up is good but again the authors have not addressed the fact that they have not followed the participants up more recently. 

Answer: Research design has been revised. As hs-CRP and ICAM-1 were not normally distributed, Wilcoxon Mann-Whitney test was applied. This text is added in data analysis section.

We do not have any data beyond 6.9 years. That is our limitation. However, a mean follow-up of 6.9 years should be sufficient time to develop MetS if depression influence MetS, according to previous studies.

Reviewer 2 Report

MetS constitutes a major problem of public health that is associated with increased risk of mortality and poor quality of life. This study examined prospectively the association between depression symptoms and metabolic syndrome or its components, and assessed the mediator role of high-sensitivity C-reactive protein and Intracellular Adhesion Molecule-1. This was an article focus on the self-reported depressive symptom and metabolic syndrome in Louisiana population, however, some confounding factors should be clarified.

  1. First of all, what is aim of this work would be liked to express? Authors should point out the relationships among the depression symptom, heart disease and metabolic syndrome biochemical parameters (i.e. lipid profiles or glucose metabolism etc.) and why such MetS parameters might exacerbate in depressed symptom.
  2. Authors should provide the known epidemiological evidence to connect the depressive status and MetS in Heart Disease population in Introduction section.
  3. This manuscript presents very limited subject number and local information. This is a general article revealed the depressive status in MetS Louisiana population but lack of novelty.

Author Response

MetS constitutes a major problem of public health that is associated with increased risk of mortality and poor quality of life. This study examined prospectively the association between depression symptoms and metabolic syndrome or its components, and assessed the mediator role of high-sensitivity C-reactive protein and Intracellular Adhesion Molecule-1. This was an article focus on the self-reported depressive symptom and metabolic syndrome in Louisiana population, however, some confounding factors should be clarified.

Answer: Several sections such as background, research design, method sections, results and conclusion were revised and highlighted in the text in yellow color.

  1. First of all, what is aim of this work would be liked to express? Authors should point out the relationships among the depression symptom, heart disease and metabolic syndrome biochemical parameters (i.e. lipid profiles or glucose metabolism etc.) and why such MetS parameters might exacerbate in depressed symptom.

Answer: In the literature, there are growing number of evidence that depression is a predictor of MetS and MetS is a risk factor for heart disease. However, the mechanism of depression leading to MetS is still not clear. Therefore, we explored the role of inflammatory biomarkers such hs-CRP and ICAM-1 in the relationship between depression and MetS. We also controlled the confounding variables such as socio-economic factors and cardiovascular risk factors at baseline.

  1. Authors should provide the known epidemiological evidence to connect the depressive status and MetS in Heart Disease population in Introduction section.

Answer:  Bogalusa Heart Study (BHS) is based on general population not from patient population.

Therefore, the BHS is unique as it is able to avoid selection bias of patient population. 

  1. This manuscript presents very limited subject number and local information. This is a general article revealed the depressive status in MetS Louisiana population but lack of novelty.

Answer: In epidemiologic study, we commonly use general population and not a particular patient population in order to avoid selection bias. Bogalusa Heart Study is an internationally recognized epidemiologic study to explore cardiovascular disease outcomes. Racial composition of Bogalusa includes White and Black populations which are unique to explore cardiovascular disease risk factors by race/ethnicity.

Local information of Bogalusa is added in the population section.

Reviewer 3 Report

The study seemed to include the interesting results, while several problems existed.

  1. The association of inflammation with systemic and psychological conditions has been known to date. The study should clearly state the novelty based on the results.
  2. The selection of biomarkers was unclear. Why did the authors select ICAM-1 among many biomarkers of inflammation?
  3. The authors used the risk ratio. The use of hazard ratio with Cox model is acceptable in general readers. Why did the authors use the risk ratio?
  4. In the method section, the representativeness of the population should be more detailed.
  5. The discrepancy of the impact on outcomes between CRP and ICAM-1 should be more prudently explained.
  6. The discussion might include the bias of rural characteristics where this population was recruited.
  7. In Table, the expression by log-added variables was not common in Journals.
  8. In Fig 3, the expression of p<0.02 was not common in Journals.
  9. In Fig 3, in theory, the line between hs-CRP (Log not needed) and abdominal obesity should be interactive.
  10. In conclusions of main text, citation of ref. 29 was not common (in the earlier discussion part, the reference should be cited and discussed).
  11. In line 141, high sensitivity hs-CRP was mistyped.
  12. In line 410 and 411, a large space appeared.
  13. In this study, if depressive subjects were found, how did the authors and research team take a care?

Author Response

Background, research design, method sections, results and conclusion sections were updated and highlighted in the text in yellow color.

  1. The association of inflammation with systemic and psychological conditions has been known to date. The study should clearly state the novelty based on the results.

Answer: We agree with you. However research is limited based on race/ethnicity. Our study was to examine racial disparity of systemic and psychological conditions with MetS in biracial (Black-White) population. 

Ref: Rohleder, Nicolas PhD Stimulation of Systemic Low-Grade Inflammation by Psychosocial Stress, Psychosomatic Medicine: April 2014 - Volume 76 - Issue 3 - p 181-189 doi: 10.1097/PSY.0000000000000049

  1. The selection of biomarkers was unclear. Why did the authors select ICAM-1 among many biomarkers of inflammation?

Answer: Inflammatory biomarkers, such as Interleukin-6, plasma Intracellular Adhesion Molecule-1 (ICAM-1) and hs-CRP promote inflammation and play a role in the pathogenesis of atherosclerosis. The epidemiological follow-up study of myocardial infarction found that both I-CAM-1 and hs-CRP were associated the CVD. Subjects with hs-CRP presented elevated coronary risk if ICAM-1 was high. Reference is added in the introduction.

Reference added as  23. Luc, G., Arveiler, D., Evans, A., Amouyel, P., Ferrieres, J., Bard, J. M., ... & PRIME Study Group. (2003). Circulating soluble adhesion molecules ICAM-1 and VCAM-1 and incident coronary heart disease: the PRIME Study. Atherosclerosis170(1), 169-176.

  1. The authors used the risk ratio. The use of hazard ratio with Cox model is acceptable in general readers. Why did the authors use the risk ratio?

Answer: This study is not time-dependable. Outcome is measured in baseline and only during the 2nd survey. Therefore hazard ratio with Cox model is not appropriate.

  1. In the method section, the representativeness of the population should be more detailed.

Answer: The method section has been revised, as suggested.

  1. The discrepancy of the impact on outcomes between CRP and ICAM-1 should be more prudently explained.

Answer: CRP and ICAM-1 were not associated MetS with this young cohort. However, only CRP was associated with central obesity, which is a component of metabolic syndrome. Although in the previous cross-sectional study, ICAM-1 was associated with metabolic component in the baseline study in total sample controlling for race, in the follow-up study, this cohort was not associated with MetS in either black or white subgroup population.

Discussion has been revised.

  1. The discussion might include the bias of rural characteristics where this population was recruited.

             Answer: SES status was added in the conclusion section.

https://worldpopulationreview.com/us-cities/bogalusa-la-population

  1. In Table, the expression by log-added variables was not common in Journals.
  2. In Fig 3, the expression of p<0.02 was not common in Journals.
  3. In Fig 3, in theory, the line between hs-CRP (Log not needed) and abdominal obesity should be interactive.
  4.  

Answer: Parameter estimate for abdominal obesity = 0.10 depression+ 0.15 log hs-CRP – 0.05 depression*Log-hs-CRP; p-value for depression 0.39, log hs-CRP 0.03 and interaction between depression*hs-CRP p=0.53 ( not included in the text). Log is removed.

  1. In conclusions of main text, citation of ref. 29 was not common (in the earlier discussion part, the reference should be cited and discussed).

Answer: Reference is added in the earlier discussion and removed ref. 29 and added common ref 3.

  1. In line 141, high sensitivity hs-CRP was mistyped.

Answer: corrected as hs-CRP

  1. In line 410 and 411, a large space appeared.

Answer: Corrected

  1. In this study, if depressive subjects were found, how did the authors and research team take a care?

     Answer:  Our findings provide support for race/ethnic-specific interventions to reduce DS or non-pharmacological effort to reduce hs-CRP to avoid the development of central obesity in White adults. A recent study showed that non-pharmacological responses such as exercise, yoga, and meditation can reduce DS without harm. Therefore, people with clinical depression should be offered either non-pharmacological or psychological interventions. A new reference is also added in the text.

Ref: O'Toole, M. S., Bovbjerg, D. H., Renna, M. E., Lekander, M., Mennin, D. S., & Zachariae, R. (2018). Effects of psychological interventions on systemic levels of inflammatory biomarkers in humans: a systematic review and meta-analysis. Brain, behavior, and immunity74, 68-78.

Round 2

Reviewer 2 Report

Authors replied the considerations properly. Please update and trim references in necessary. I have no further questions. 

Author Response

"Answer: References are updated and trimmed to #35 from #45. We improved the writing section in the research design, methods, results and conclusion section. We highlighted in tracking and yellow color.

 Thank you so much for your valuable comment and suggestion. We appreciate it."

Reviewer 3 Report

Character history of correction of the words and numerical values remains in the revised version of text. We could not read overall corrected parts.

For increasing the study’s values, the following points can be addressed again.

Did triglyceride values have a normal distribution?

In the first part of discussion, the novelty of the study may be more stressed.

In conclusions, the references were cited, which is not common. The sentences with references may be discussed before conclusions.

Author Response

As reviewer indicated moderate English changes required, manuscript is re-evaluated by professional editor and author and marked by color and tracking. Thus, it will help improving method, results and conclusion sections.

Did triglyceride values have a normal distribution?

Answer: I checked it. It is not normally distributed. I reported their median values along  25th and 75th percentile. I also changed values of insulin and glucose as they are also not normally distributed.

However, in multivariate model, we did log transformation of non-normally distributed variables before putting in the model.

In the first part of discussion, the novelty of the study may be more stressed.

Answer: I changed the 1st part of discussion section and highlighted in yellow color

In conclusions, the references were cited, which is not common. The sentences with references may be discussed before conclusions.

Answer: I changed the conclusion part and removed those references.

Thank you so much for your feedback, comment and suggestion. We appreciate your feedback.

This manuscript is a resubmission of an earlier submission. The following is a list of the peer review reports and author responses from that submission.

Round 1

Reviewer 1 Report

This is an informative attempt at investigating the association between depression symptoms and metabolic syndrome (MetS) and the mediating role of high-sensitivity C-reactive protein (hs-CRP). The manuscript is fairly well organised and written. The manuscript may be improved with attention to several specific details:

Minor:
1) Define all abbreviations when they are first use, including in the abstract and introduction (e.g., HDL-C and BP)
2) Line 45: Which population? I assume the US but please specify
3) Please change all instances of ‘Blacks’ and ‘Whites’ to Black individuals and White individuals to prevent being reductionist based on ethnicity.
4) The term ‘biracial asymptomatic young adults’ (lines 69-70) is misleading. The individuals were not biracial although the sample was. Is that correct? Either be more precise here by saying “…during a follow-up period of 6.9 years in 419 White and 180 Black asymptomatic young adults” or leave race/ethnicity information out of this sentence and specify more clearly in the Method section. In addition, I was wondering, and it may be worth specifying, were any of the individuals actually biracial or did they all identify as either Black or White?
5) What are “General Measures”? (line 87) “Trained field observers used standardized techniques and protocols.” (line 88) To measure what exactly? Please be more specific.
6) “The lifestyles variables, such as smoking and alcohol consumption, were also assessed during the survey.” (lines 91-92). Could the authors please provide more information on this survey? What was it asking? How long was it? Was it in person? Online? Please be more specific or else provide a reference to a publication describing the survey.
7) “DS was assessed using the Epidemiological Studies-Depression scale (CES-D) developed by Laurie Radloff in 19” (line94-95) It looks like the rest of the year is missing.
8) Please provide more information on when and how serum was collected.
9) “Drinking alcohol, CVD risk factors, and hs-CRP were substantial predictors of developing MetS in Blacks” (line 164-165) but the p-value for alcohol in table 2 is 0.08 and the p-value for smoking is 0.02.
10) Can you please comment on why DS is not a significant predictor of MetS in either race in your first model (line 166) but then in the crude model in Table 3 it is a significant predictor. In line with this, I find the result section hard to follow. Is there a way to present the results more easily?

Major:
1) Race/ethnicity is introduced early in the introduction although no attempt is made to explain the seemingly discrepancies between previous research and the hypothesis of the here presented research, i.e., “MetS has a high prevalence in Blacks, especially in Black women and that is attributable to the disproportionate occurrence of the higher incidence of obesity, hypertension, and diabetes among Blacks.” (line 45-47), “Depression is more common in Whites than in Blacks” (line 51), “Blacks have higher hs-CRP compared to Whites” (line 63) and “We hypothesized that depression symptoms influence metabolic syndrome or its components through inflammatory biomarker measured by hs-CRP.” (lines 66-67). What the introduction is missing is a more balanced presentation of previous research. Please explain all of the following within your manuscript: Why is it important to introduce race/ethnicity so early on? What is known about the aetiology and prevalence of MetS in Black and White individuals? Similarly, what is known about the prevalence of depression and its link to MetS in Black and white individuals? What do we know about any potential difference in hs-CRP between Black and White individuals, and the respective links between hs-CRP to MetS or depression?
2) The authors have not made it clear to the reader why they think that the direction of the effect is from depression to inflammation to MetS and not the other way around. Please comment on that.
3) Why was high-sensitivity C-reactive protein rather than other inflammatory markers used?
4) Could the authors please comment on whether an attempt has been made to follow-up the participants of this study, given that the data are from 2008-9?

Author Response

Point-by-Point Responses

Review 1

Comments and Suggestions for Authors

Minor:
1) Define all abbreviations when they are first use, including in the abstract and introduction (e.g., HDL-C and BP)

Response: The abbreviations are defined.
2) Line 45: Which population? I assume the US but please specify

Response: Corrected as US population.

3) Please change all instances of ‘Blacks’ and ‘Whites’ to Black individuals and White individuals to prevent being reductionist based on ethnicity.

Response: Blacks and Whites are changed to Black individuals and White individuals respectively.

4) The term ‘biracial asymptomatic young adults’ (lines 69-70) is misleading. The individuals were not biracial although the sample was. Is that correct? Either be more precise here by saying “…during a follow-up period of 6.9 years in 419 White and 180 Black asymptomatic young adults” or leave race/ethnicity information out of this sentence and specify more clearly in the Method section. In addition, I was wondering, and it may be worth specifying, were any of the individuals actually biracial or did they all identify as either Black or White?

Response: Removed the term “biracial”.  Those sentences are modified accordingly.  

5) What are “General Measures”? (line 87) “Trained field observers used standardized techniques and protocols.” (line 88) To measure what exactly? Please be more specific.

Response:  Trained field observers used standardized techniques and protocols to measure anthropometric and blood pressure measurements. Revised the method.

6) “The lifestyles variables, such as smoking and alcohol consumption, were also assessed during the survey.” (lines 91-92). Could the authors please provide more information on this survey? What was it asking? How long was it? Was it in person? Online? Please be more specific or else provide a reference to a publication describing the survey.

Response: The lifestyles variables, such as smoking habit and alcohol consumption, were also assessed during the survey in person. The following reference is added:

Berenson, G.S.; Srinivasan, S.R.; Bao, W.; Newman, W.P.; Tracy, R.E.; Wattigney, W.A. Association between multiple cardiovascular risk factors and atherosclerosis in children and young adults. The Bogalusa Heart Study. N Engl J Med, 1998, 338, 1650– 6.

7) “DS was assessed using the Epidemiological Studies-Depression scale (CES-D) developed by Laurie Radloff in 19” (line94-95). It looks like the rest of the year is missing.

Response: Corrected as Laurie Radloff (1977).

8) Please provide more information on when and how serum was collected.

Response: Participants were asked to fast for 12 hours before coming to the BHS center for screening. The compliance was checked by trained field officer during interview on the morning of screening. We added this information in the method section.

9) “Drinking alcohol, CVD risk factors, and hs-CRP were substantial predictors of developing MetS in Blacks” (line 164-165) but the p-value for alcohol in table 2 is 0.08 and the p-value for smoking is 0.02.

Response: “drinking alcohol” variable is removed from the text. Most of CVD risk factors and hs-CRP were substantial predictors of developing MetS in Black participants.

10) Can you please comment on why DS is not a significant predictor of MetS in either race in your first model (line 166) but then in the crude model in Table 3 it is a significant predictor. In line with this, I find the result section hard to follow. Is there a way to present the results more easily?

Response:  We presented two new graphs in the manuscript.

Figure 1: Incidence of MetS and its component by DS among White Participants

It showed that there was no development of MetS and its of component by DS status at baseline. Only central obesity measured by waist circumference was significantly associated with DS (p <0.02). MetS and all other components were not at significant (p > 0.05).

Figure 2: Incidence of MetS and its component by DS among Black Participants

Figure 2 displayed that the incidence of MetS and its of component by DS status at baseline. There was no incidence of MetS and its component was associated with DS among Black participants (p >0.05).

Response: Please note that Table 3 Incidence of MetS and Table 4 for Component of MetS, central obesity. Now we displayed with graph as per your recommendation.

Major:
1) Race/ethnicity is introduced early in the introduction although no attempt is made to explain the seemingly discrepancies between previous research and the hypothesis of the here presented research, i.e., “MetS has a high prevalence in Blacks, especially in Black women and that is attributable to the disproportionate occurrence of the higher incidence of obesity, hypertension, and diabetes among Blacks.” (line 45-47), “Depression is more common in Whites than in Blacks” (line 51), “Blacks have higher hs-CRP compared to Whites” (line 63) and “We hypothesized that depression symptoms influence metabolic syndrome or its components through inflammatory biomarker measured by hs-CRP.” (lines 66-67). What the introduction is missing is a more balanced presentation of previous research.

 Please explain all of the following within your manuscript: Why is it important to introduce race/ethnicity so early on? What is known about the aetiology and prevalence of MetS in Black and White individuals? Similarly, what is known about the prevalence of depression and its link to MetS in Black and white individuals? What do we know about any potential difference in hs-CRP between Black and White individuals, and the respective links between hs-CRP to MetS or depression?

Response: The introduction section has been modified, as suggested.
2) The authors have not made it clear to the reader why they think that the direction of the effect is from depression to inflammation to MetS and not the other way around. Please comment on that.

Response: The revised introduction section addresses the suggestions.

3) Why was high-sensitivity C-reactive protein rather than other inflammatory markers used?

Response: High-sensitivity C-reactive protein and its pre-cursor IL-6 are widely used in behavioral research for indicating how social and behavioral variables effect on health. Numerous research focus on mediating role of chronic low grade inflammation mechanism. These are putative inflammatory biomarkers and commonly assayed biomarkers to reflect the underlying inflammatory conditions.

4) Could the authors please comment on whether an attempt has been made to follow-up the participants of this study, given that the data are from 2008-9?

Response: BHS is designed to investigate the cardiovascular risk factors using both cross-section and longitudinal studies in nature. Study participants were tracking from childhood to adulthood since 1973. In 2001-2002, 1203 young adults participated for aging study. In 2008-2009, 914 young adults were studies. After merging two survey data, 797 participants were eligible to this study.

Reviewer 2 Report

This is an epidemiological biracial study, analyzing the interaction of depression system and it's influence on development & progression of metabolic syndrome using established criteria. Study finds blacks having higher BMI and waist circumference compared to white counterparts. However LDL and triglycerides (risk factor for cardiovascular insult) were lower in blacks compared to whites.  Since depression system seems to influence metabolic syndrome through inflammation, an increased CRP level is seen because CRP level is upregulated during proinflammatory environment.

 There is no data for level of inflammatory cytokines (IL-6, IL-1). Mere presence of CRP cannot be attributed as mediator for depression and metabolic syndrome. Accordingly, in the manuscript text ; mediator role of CRP should be changed to increased level of CRP.

Author Response

Reviewer 2

This is an epidemiological biracial study, analyzing the interaction of depression system and it's influence on development & progression of metabolic syndrome using established criteria. Study finds blacks having higher BMI and waist circumference compared to white counterparts. However LDL and triglycerides (risk factor for cardiovascular insult) were lower in blacks compared to whites.  Since depression system seems to influence metabolic syndrome through inflammation, an increased CRP level is seen because CRP level is upregulated during proinflammatory environment.

There is no data for level of inflammatory cytokines (IL-6, IL-1). Mere presence of CRP cannot be attributed as mediator for depression and metabolic syndrome. Accordingly, in the manuscript text ; mediator role of CRP should be changed to increased level of CRP.

Response:

Obesity play role in systemic low grade inflammation. Adipose tissue releases pro inflammatory cytokines like IL-1 and IL-6.  Hs-CRP is released by the liver in response to IL-6. Both act as a marker and promoter of atherogenesis.

The recent study showed that these cytokines regulate the proliferation and apoptosis of adipocytes, promote lipolysis, inhibit lipid synthesis and decrease blood lipids through autocrine and paracrine mechanisms. (Ref Wang T,  He C. (2018). Pro-inflammatory cytokines: The link between obesity and osteoarthritis.

Reviewer 3 Report

This is my opinion on the article : "Progression of Metabolic Syndrome Component along with Depression Symptoms and High Sensitivity C-Reactive Protein: The Bogalusa Heart Study".

The abstract already shows what the result of the work will be and already raises doubts.   The materials and methods are suitable for describing the work.   The data can be traced back to more 12 years ago; the big problem is that the references are almost all dated in the decade 2000-2010. Although the data relating to 2008, the conclusions should be based on ongoing work and future prospects (the conclusions are unsatisfactory and hasty).

I think there are some aspects to be clarified:

  • Why in Blancks there was no evidence of the mediator role of sh-CRP on abdominal obesity, it can be linked to a small number of participants?
  • DS influences central obesity (Fig.1). Does central obesity cause DS?
  • Your results provided evidence of DS effects on central obesity and also highlighted the mediator role of hs-CRP. Have you considered the inverse relationship?

Author Response

Reviewer 3

This is my opinion on the article : "Progression of Metabolic Syndrome Component along with Depression Symptoms and High Sensitivity C-Reactive Protein: The Bogalusa Heart Study".

The abstract already shows what the result of the work will be and already raises doubts.   The materials and methods are suitable for describing the work.   The data can be traced back to more 12 years ago; the big problem is that the references are almost all dated in the decade 2000-2010. Although the data relating to 2008, the conclusions should be based on ongoing work and future prospects (the conclusions are unsatisfactory and hasty).

Response: We agree that the data are old. However, BHS data are tracked back to 1973 and widely used to see how childhood factors are risk factor for CVD. BHS also used to track the childhood risk factor and how they correlated with adults’ risk factors. Based on your suggestions, the reference lists and conclusion section are modified.

I think there are some aspects to be clarified:

  • Why in Blanks there was no evidence of the mediator role of hs-CRP on abdominal obesity, it can be linked to a small number of participants?

Response: Yes, we don’t have enough power to detect mediator role of hs-CRP on abdominal obesity. This has been mentioned in the revised manuscript.

  • DS influences central obesity (Fig.1). Does central obesity cause DS?
    Response: We do not have data on DS as outcome variable. Reverse causality cannot be established.
  • Your results provided evidence of DS effects on central obesity and also highlighted the mediator role of hs-CRP. Have you considered the inverse relationship?

Response: We do not have data on DS as outcome variables. DS is measured only at baseline.

Round 2

Reviewer 1 Report

I thank the authors for addressing all of the minor points I had and I can see that they have put in a large amount of effort to revise this manuscript. However, I do not feel that they have addressed my more serious concerns with this paper. 

The authors did not answer my question as to whether an attempt has been made to follow up this sample since 2009. While a lot of research has used high-sensitivity C-reactive protein and IL-6, the research community is slowly moving away from using single inflammatory markers and moving towards cytokine panels as they are more informative particularly of low grade chronic inflammation. 

Lastly, I personally don't believe that the direction of effect is from depression to inflammation to MetS and the authors have not been able to convince me otherwise with their introduction or their study. I therefore think the premise of the study may be flawed. 

Reviewer 3 Report

In my opinion some aspects are needed to be clarified. Although the authors says to specify some limitations in the revised manuscript, I think that it is not suitable for the publication.